# The Contribution of *Legionella anisa* to *Legionella* Contamination of Water in the Built Environment

**DOI:** 10.3390/ijerph21081101

**Published:** 2024-08-20

**Authors:** Brian Crook, Charlotte Young, Ceri Rideout, Duncan Smith

**Affiliations:** 1Science and Research Centre, Health and Safety Executive, Harpur Hill, Buxton, Derbyshire SK17 9JN, UK; 2Specialist Division Occupational Hygiene, Health and Safety Executive, Cardiff CF10 1EP, UK; 3Specialist Division Health Unit, Health and Safety Executive, Newcastle upon Tyne NE98 1YX, UK; duncan.smith@hse.gov.uk

**Keywords:** *Legionella pneumophila*, *Legionella anisa*, water systems, hospital, disinfection and recolonization, COVID-19 lockdown

## Abstract

*Legionella* bacteria can proliferate in poorly maintained water systems, posing risks to users. All *Legionella* species are potentially pathogenic, but *Legionella pneumophila* (*L. pneumophila*) is usually the primary focus of testing. However, *Legionella anisa* (*L. anisa*) also colonizes water distribution systems, is frequently found with *L. pneumophila*, and could be a good indicator for increased risk of nosocomial infection. Anonymized data from three commercial *Legionella* testing laboratories afforded an analysis of 565,750 water samples. The data covered July 2019 to August 2021, including the COVID-19 pandemic. The results confirmed that *L. anisa* commonly colonizes water distribution systems, being the most frequently identified non-*L. pneumophila* species. The proportions of *L. anisa* and *L. pneumophila* generally remained similar, but increases in *L. pneumophila* during COVID-19 lockdown suggest static water supplies might favor its growth. Disinfection of hospital water systems was effective, but re-colonization did occur, appearing to favor *L. pneumophila*; however, *L. anisa* colony numbers also increased as a proportion of the total. While *L. pneumophila* remains the main species of concern as a risk to human health, *L. anisa*’s role should not be underestimated, either as a potential infection risk or as an indicator of the need to intervene to control *Legionella*’s colonization of water supplies.

## 1. Introduction

Main water supplies are intended to be wholesome and safe for drinking, cooking, and bathing, but they are not sterile and can contain pathogenic bacteria. If hot- and cold-water systems (HCWSs) are not well designed or managed, then these waterborne pathogens, including *Legionella* bacteria, can proliferate to the extent that they pose risks to users of the water system [1]. Controlling the growth of legionellae in large buildings with complex water systems can be challenging. This is especially the case if it is exacerbated by poor designs that make maintaining safe water temperatures problematic, or building modifications or construction materials that encourage the development of biofilms and provide pockets within the water system where legionellae can thrive [1]. 

At least half of all known *Legionella* species have been shown to cause human disease, but all species are thought to be potentially pathogenic [2]. The major clinical manifestation of infection due to *Legionella* species is a type of pneumonia known as Legionnaires’ disease (LD); however, non-pneumonic legionellosis (Pontiac Fever) and extrapulmonary infection may occur [2]. *Legionella pneumophila* (*L. pneumophila*) is the most commonly detected species amongst clinical cases of legionellosis. For example, in the USA, more than 90% of isolates are *L. pneumophila*, and 83% are *L. pneumophila* serogroup 1 (Sg1) [3], while in the EU, 83% of isolates were *L. pneumophila* Sg1 in 2020, and 82% in 2021 [4]. While *L. pneumophila* is the species most frequently cultured from the natural environment, other disease-relevant *Legionella* species, such as *L. micdadei*, *L. bozemanii*, *L. longbeachae*, and *L. anisa*, have yet to be extensively explored in premise plumbing systems [5], despite the fact that *L. anisa* was first isolated from environmental samples of cooling-tower and potable water sources as long ago as 1980 [6]. *Legionella* species other than *L. pneumophila* have been reported to cause disease but are less well researched [7,8,9]. Non-*L. pneumophila* species occasionally cause LD [8,10,11], severe pneumonia [12], and Pontiac Fever [13]. In total, 2–7% of the cases of legionelloses are estimated to be caused by non-*L. pneumophila* species [14], with most of these confirmed infections occurring in immunosuppressed patients [2]. Non-*L. pneumophila* species are less commonly detected in clinical diagnoses or during environmental surveillance due to their slow culture growth and the absence of specific and rapid diagnostic/analytical tools [15].

LD outbreaks are more commonly associated with HCWSs than with other system types; however, this may reflect the higher number of them compared to other water systems in the built environment [16]. As part of an overall water management strategy, periodic sampling and testing for the presence of legionellae in water systems can be extremely valuable in determining whether the microbiological risk controls applied are effective [1]. For the testing of environmental samples, agar plate culture is still the most widely used standard technique for the detection and quantification of viable *Legionella* spp. [17,18]; however, this technique does favor *L. pneumophila* over non-pneumophila species [19].

While *L. pneumophila* is usually the primary focus of testing, *L. anisa* is also a common colonizer of water distribution systems, even though it is an uncommon hospital-acquired pathogen [20]. Table 1 summarizes evidence of the potential contribution of *L. anisa* to the microbiome of water systems in the built environment and its potential to cause human disease. In brief, studies published previously have shown the potential for *L. anisa* to be a colonizer of water distribution systems at a comparable level to *L. pneumophila*, as well as for *L. anisa* to be responsible for a range of pneumonic and non-pneumonic infections. While this adds to the importance of recognizing the implications of the presence of *L. anisa* in water distribution systems, other studies have shown that conventional clinical tests for legionellosis, such as urinary antigen tests (UATs), may underestimate the role of non-*L. pneumophila* species. However, in countries where clinical laboratories have adopted polymerase chain reaction (PCR)-based test methods, the role of non-*L. pneumophila* species, including *L. anisa*, in reported respiratory infections has been recognized.

This may suggest that *L. anisa* is less pathogenic to humans than *L. pneumophila* [21,22]; however, there is also likely to be significant under-reporting of clinical cases due to the diagnostic testing undertaken.

This is supported by data from the UKHSA Legionella Reference Laboratory. In an examination of isolates retained by the laboratory from environmental and clinical samples submitted, and thus associated with clinical infection, over a period from 2004 to 2021, from a total of 2321 environmental samples, we found that 2179 (93.9%) were *L. pneumophila* and 96 (4.14%) were *L. anisa*; and, during the same period, from a total of 1360 clinical samples, 1327 (97.6%) were *L. pneumophila* and only 1 (0.07%) was *L. anisa* (V. Chalker, UKHSA, pers comm.).

Culture is still the primary method used to detect legionellae in environmental samples, but the identification of isolates has been revolutionized by the application of Matrix-assisted laser desorption ionization–time-of-flight mass spectrometry (MALDI-ToF MS). This widely accepted method to identify bacterial species grown on agar plates can be applied to legionellae [13,27]. It accurately identifies a range of bacteria with a high positive predictive value from 95.4% [36] to 99.2% [37], and for *Legionella*, MALDI-ToF MS has shown a 99.1% specificity [38].

Other detection methods, for example, Lateral Flow or Most Probable Number methods, may compare favorably with culture for the detection of *L. pneumophila*; however, they do not detect non-*L. pneumophila* species, including *L. anisa* [1,44].

Real-time and quantitative PCR methods have been developed for *Legionella* species [16], and multiplex PCR assays have been developed which allow for the simultaneous detection of *Legionella* species and *L. anisa* [5].

Regarding the value of responding to an identified problem in the built environment from legionellae other than *L. pneumophila*, UK health and safety guidance [45] suggests that their presence or growth in a water system could indicate that suitable growth conditions for *L. pneumophila* have, at some point, been achieved. As such, these organisms should be seen as indicators of system colonization, and appropriate action should be taken upon their detection [17,23].

*L. anisa* is frequently found in conjunction with *L. pneumophila* in hospital water systems [46,47,48] and could be a good indicator for increased risk of nosocomial LD [23]. It has also been suggested that the presence of a more abundant *Legionella* population, including *L. anisa*, in a water system may limit the detection of *L. pneumophila*; therefore, action should be taken to eradicate all *Legionella* contamination of hospital water distribution systems [2,23].

*L. anisa* in biofilm in hot-water systems can be difficult to remove via thermal disinfection [49] and may also show some degree of biocidal resistance in hot water-system biofilm when compared to *L. pneumophila* [28,50]. *L. anisa* concentrations in water systems can be reduced using chlorine dioxide disinfectant; however, this can be a protracted process [51].

By monitoring both *L. pneumophila* and non-*L. pneumophila* species, changes to building management can be made to address potentially different risks at different times of the year. Higher concentrations of *Legionella* spp. may occur in buildings and hot-water taps during periods of lower water use [13]. Poor water circulation and temporary or permanent water stagnation allow *Legionella* to colonize water systems [52]. Consequently, building shutdowns or reduced occupancy during the COVID-19 pandemic could have increased colonization, and alerts were issued to address this risk [53]. Interestingly, when examining a subset of the UKHSA Reference Laboratory data over a later time frame of June 2019 to July 2021, thus spanning the Covid-19 lockdown period in the UK, we noted that, out of a total of 393 environmental samples, 359 (91.35%) were *L. pneumophila* and 31 (7.89%) were *L. anisa*; and, during the same period, from a total of 204 clinical samples, 194 (95.1%) were *L. pneumophila* but none (0.0%) was *L. anisa* (V. Chalker, UKHSA, pers comm.).

This paper describes an analysis of data on routine water testing for *Legionella* species undertaken in the UK by commercial laboratories over a period from July 2019 to August 2021. The primary aim of this study was to examine the relationship between the presence of *L. anisa* and *L. pneumophila* in water samples submitted for routine testing, including before and after a clean-up of a hot- and cold-water systems. In addition, as this period spanned the pandemic, and the requirement for premises to shut down, it was possible to examine Legionella monitoring data before and after shutdown.

## 2. Materials and Methods

The Legionella Control Association is a UK trade body with which one of the authors (D.S.) works closely. Through this connection, contacts were made with commercial laboratories that undertake routine water testing for *Legionella* for a range of clients. Samples are likely to be taken for a variety of reasons, for example, routine monitoring as a check on control effectiveness (as per national guidance recommendations) [54], incident investigation, or risk assessment. Water samples from premises were regularly and routinely collected from predetermined sampling points, following the British Standard Code of Practice [55], and delivered to the testing laboratory. Here, they followed the standard ISO protocol [56] to test for *Legionella*. In brief, water samples were concentrated by membrane filtration, or directly plated or diluted as appropriate; heat- and/or acid-treated to reduce non-target bacteria; and then used to inoculate buffered charcoal yeast extract agar plates with appropriate supplements. Total numbers of Legionella in samples are reported back to clients as colony-forming units (CFUs) per L of water, and suspect *Legionella* colonies are identified by MALDI-ToF MS.

Three laboratories agreed to share anonymized data, each providing a database of test results with numbers of samples testing positive for *L. pneumophila*, *L. anisa*, and a range of other *Legionella* species.

## 3. Results

### 3.1. Overview of Data Sets Provided for Analysis and Limitations of the Data Obtained

The three *Legionella* testing laboratories that agreed to participate voluntarily provided data from their routine testing of water samples for *Legionella*, as undertaken for a range of clients. This totaled over 560,000 data points across the three data sets, thus enabling a valuable “bigger picture” analysis to be undertaken. All samples were collected using British Standard methodology [55] and were analyzed by each laboratory using ISO standardized techniques [56]. For each laboratory, the standard protocol was to take representative isolates of suspect *Legionella*-positive colonies and undertake a MALDI-ToF MS analysis to confirm the species level and, for *L. pneumophila*, also determine the serogroup level.

There were, however, some limitations to the information supplied, as follows. As these data were from commercial clients, it was not appropriate for them to supply details such as sample location or industry type, nor were details such as time of sampling available. The exception to this was Data Set 2, for which samples were taken from a hospital as a result of *Legionella* colonization being identified. The information supplied therefore was of *Legionella* testing before and after a cleaning intervention. This did provide sample details, such as whether the sample was taken from a shower hose or a thermostatic mixing valve tap. Although not all sample locations were sampled both pre- and post-cleaning, the overall effects of the intervention could be analyzed.

There were some additional differences in how each laboratory supplied its data. One data set provided information on *Legionella*-positive samples only, while the other two provided information on the total number of samples tested, i.e., both *Legionella*-positive and no *Legionella* detected. In two of the three data sets, the *Legionella*-positive sample data were also broken down by month, one being from July 2019 to August 2021 and the other from January 2020 to June 2021. For the third data set, the results provided were not time-resolved but spanned a similar time frame, from 2019 to 2021. Although each data set provided a breakdown of *Legionella*-positive results into a broad range of species, for the purpose of the current analysis, the focus was on *L. pneumophila* and *L. anisa*; therefore, other species were grouped together as “other”.

### 3.2. Legionella Data Set 1

This data set provided information on *Legionella*-positive samples only. Between July 2019 and August 2021, nearly 12,000 positive *Legionella* samples were recorded. As summarized in Appendix A (available online), the most prevalent species identified in these samples was *L. pneumophila* (47%). The next most prevalent species identified was *L. anisa* (46%). These data are presented in Figure 1 as monthly data over the 26-month period.

Over the 26-month period, the general proportion of *L. anisa* and *L. pneumophila* remained similar. However, there were some months with large differences, most noticeably April and May 2020 and June 2021.

Expressed as a percentage of the total number of *Legionella*-positive samples, those identified as *L. pneumophila* ranged from 35% to 70%, with April and May 2020 having the largest percentage of the total, at 63% and 70%, respectively, which also corresponded with a decrease in the percentage of samples identified as *L. anisa* (Appendix A). For *L. anisa*, the overall percentage of the total was generally slightly lower than for *L. pneumophila* and ranged from 23% to 59%. The 23% value in June 2021 corresponded to a month when the percentage of other *Legionella* species was greatest, at 26%, seen clearly in Figure 2, which was an outlier from the proportion of other *Legionella* species usually found.

By March 2020, as the COVID-19 pandemic started to have global impact, lockdown restrictions were applied in the UK leading to reduced building occupancy [57]. This may have influenced the number of samples taken and the number of *Legionella*-positive samples recorded. In Figure 1, an arrow indicates when the UK COVID-19 lockdown began, more specifically from 23 March 2020. Initially the number of *Legionella*-positive samples fluctuated between April 2020 and June 2020, before increasing to its highest in July 2020. This may be due to places of employment being closed for a 3-month period, with stagnation of water supplies, before gradually being reopened from late June 2020 onwards. Seasonal fluctuations in positive samples are another potential influence, with higher numbers of positives in the warmer summer/early autumn period. This may be evident to some extent in August 2019 but more evident in July/August 2020 and July 2021, and in the post-lockdown period, May to December 2020, where each month had higher numbers of *Legionella*-positive samples than in any pre-COVID lockdown months in the data set.

Overall, 23% of *Legionella*-positive samples were taken in the eight months of the data set pre-COVID and 77% in the 18 months post-COVID. However, the proportion of those samples identified as *L. anisa* did not alter (approximately 46% of *Legionella*-positive samples; Appendix A). A Chi-square test indicated that there is no significant association between *Legionella* species and pre/post-COVID (*p* = 0.467).

### 3.3. Legionella Data Set 2

This data set related to an intervention undertaken at a hospital between January 2020 and June 2021 after routine monitoring identified significant levels of *Legionella*. A site-wide disinfection of all hot- and cold-water systems was undertaken using hydrogen peroxide, with samples retaken 2-to-7 days post-disinfection, as per guidance to ensure no residual disinfectant interferes with microbiological results [45]. An intensive post-cleaning and disinfection sampling exercise was undertaken to assess the effectiveness of the intervention. This data set included samples taken from shower hoses, cold-water taps, or handwashing basins with taps incorporating thermostatic mixing valves (TMVs). The data were presented in terms of the presence or absence of colonies grown on selective isolation plates, indicative of the number *Legionella*-positive samples before and after disinfection. Additional information provided in this data set was of a species breakdown, following MALDI-ToF MS analysis, of numbers of *Legionella*-positive colonies per liter of water sampled, and for some data analyses, this species breakdown was aggregated into the total numbers of *L. pneumophila* or *L. anisa* colonies that were isolated. Out of a total of 613 samples taken, 322 were positive for *Legionella* (52.5%). Overall, 167 of these were pre-cleaning samples (out of 283; 59.0%), and 155 were post-cleaning samples (out of 330; 47.0%). The most prevalent species identified in these samples was *L. pneumophila* (49%). The next most prevalent species identified was *L. anisa* (38%). Figure 3 shows the total number of *Legionella*-positive samples, by species, before and after the cleaning intervention.

Figure 4 shows a monthly breakdown of the data. Although the data span an 18-month period, the sampling was not undertaken every month; thus, data are missing for April 2020, August to October 2020, and January 2021, and no pre-cleaning samples were taken for the first six months (see Appendix A). Due to relatively low numbers of some sampling results, a statistical analysis was not feasible.

Only post-cleaning samples are available that span the two time frames of pre- and post-COVID. Overall, 17% of *Legionella*-positive samples were taken pre-COVID, and 83% post-COVID. The proportion of those samples identified as *L. anisa* roughly halved post-COVID compared to pre-COVID (from 65% of *Legionella*-positive samples down to 33%). See Figure 5.

Table 2 presents data where there were both pre- and post-cleaning samples (n = 282). The breakdown of species identified in *Legionella*-positive sample pre-cleaning is presented across the rows, and for post-cleaning, it is presented in columns. For example, out of 61 locations, at 18, *L. anisa* remained the predominant species identified post-cleaning, 6 changed to having *L. pneumophila* as the predominant species, and three changed to other *Legionella*, with 34 locations going from *L. anisa* being isolated to none being detected. For *L. pneumophila*, at 54 out of 81 locations, *L. pneumophila* remained the predominant species identified, 8 changed to having *L. anisa* as the predominant species, and four changed to other *Legionella*, with 15 locations going from *L. pneumophila* being isolated to none being detected. This meant that *L. anisa* went from predominating at 61 locations pre-cleaning to 38 locations post-cleaning, and *L. pneumophila* went from predominating at 81 locations pre-cleaning to 71 locations post-cleaning. Locations where no Legionellae were detected increased from 115 pre-cleaning to 161 post-cleaning.

In addition to the number of *Legionella*-positive samples, this data set also included *Legionella*-positive colony count data. These were obtained by the testing laboratory by recording colony counts and then by taking representative isolates and identifying them using MALDI-ToF MS. Between January 2020 and June 2021, there were over 630,000 positive *Legionella* colonies recorded. Approximately 70% of these were from pre-cleaning samples, and 30% were from post-cleaning samples. The most prevalent species identified in these colonies was *L. pneumophila* (70%). The next most prevalent species identified was *L. anisa* (20%). See Table 3.

As also seen from Table 3, for pre-cleaning, *L. anisa* made up 17.2% of the total, and *L. pneumophila* made up 73.8%. For post-cleaning, of a much lower number of colonies (193,869 compared to 437,204), *L. anisa* made up a larger proportion, at 26.4%, up from 17.2% of the total, and the proportion of *L. pneumophila* was reduced to 64.1%, down from 73.8%. Out of 282 data points that had pre- and post-cleaning colony counts, counts went down in 136 (48.2%) and were unchanged from none being detected in 103 (36.5%) instances. In only 43 (5.2%) instances were colony counts found to increase post-cleaning. The most notable of instances were a shower hose that went from a pre-clean colony count of 842 CFU/L *L. pneumophila* Sg1 to 4400 CFU/L, comprising 400 *L. pneumophila* Sg1 and 4000 *L. anisa*; a shower hose that went from a pre-clean colony count of 779 CFU/L *L. pneumophila* Sg1 to 8667 CFU/L *L. pneumophila* Sg1; a cold tap that went from a pre-clean colony count of 1200 CFU/L *L. anisa* to 5000 CFU/L *L. pneumophila* Sg1; and a shower hose that went from none detected pre-clean to a colony count of 5400 CFU/L *L. anisa*. Of the 136 data points showing a reduction in colony counts from pre- to post-cleaning, 19 showed a reduction by >5000 CFU/L. The most notable cases are as follows: a shower hose that went from a pre-clean colony count of 16,421 CFU/L *L. pneumophila* Sg1 to 489 CFU/L *L. pneumophila* Sg1; a shower hose that went from a pre-clean colony count of 12,600 CFU/L *L. pneumophila* Sg1 to 200 CFU/L *L. pneumophila* Sg1; a shower hose that went from a pre-clean colony count of 12,400 CFU/L *L. pneumophila* Sg1 to 100 CFU/L *L. pneumophila* Sg1; a shower hose that went from a pre-clean colony count of 20,000 CFU/L *L. pneumophila* Sg1 to 4600 CFU/L *L. pneumophila* Sg1; a shower hose that went from a pre-clean colony count of 44,200 CFU/L *L. pneumophila* Sg1 to 4400 CFU/L *L. pneumophila* Sg1; and a shower hose that went from a pre-clean colony count of 20,600 CFU/L *L. pneumophila* Sg1 to 3200 CFU/L *L. pneumophila* Sg1. Although not statistically proven, the above data indicated that the highest counts were greatly reduced by the hydrogen peroxide-disinfection intervention.

Most of the sample points were either showers or handwashing basins with taps incorporating TMVs. The data in Appendix A summarize how colony counts and the species isolated compare for both of these sample types pre- and post-cleaning, and this is expressed graphically in Sankey diagrams in Figure 6 for shower hoses and in Figure 7 for TMVs to show species changes. The purpose of a Sankey diagram is to provide a visual representation of change from one state to another, with the width of the arrows proportional to the flow rate. In this instance, they were used to visualize the distribution and change in the presence of *Legionella* pre- and post-cleaning for showers and TMV taps.

A greater proportion of TMVs than showers showed no presence of *Legionella* both pre- and post-cleaning (35.4% compared to 14.9%), and also more changed from having *L. pneumophila* Sg1 colony counts/L pre-clean to none detected post-clean (8.1% compared to 1.4%). The proportion that changed from *L. anisa* colony counts to none detected was similar for both sample types. Although *Legionella* colony counts were not being eliminated, the effect of the cleaning intervention on reducing colony counts appeared to be greater with showers—*L. pneumophila* Sg1 and *L. anisa* colony counts/L were reduced in 23% and 12.2% of samples, respectively, compared to 7.4% and 1.9%, respectively, in TMVs. Although there were only a few instances where colony counts/L were increased post-cleaning compared to the same sample point pre-cleaning, there was some indication that showers were more likely to be recolonized by either species. In 5.4% of shower samples, *L. pneumophila* Sg1 remained the predominant colonizing species, with counts/L increased, compared to 3.1% of TMVs. Also, in 5.4% of shower samples, *L. anisa* remained the predominant colonizing species, with counts/L increased, compared to 1.2% of TMVs, and 4.1% of shower samples changed from none detected to *L. anisa* pre- to post-clean compared to 1.9% of TMVs, also suggesting the possibility that *L. anisa* had a greater propensity for recolonization.

Only post-cleaning samples were available that span the two time frames, pre- and post-COVID. Overall, 13% of *Legionella*-positive sample colonies were identified pre-COVID, and 87% post-COVID. A majority of colonies identified pre-COVID were *L. anisa* (77%); however, post-COVID, a majority of colonies were *L. pneumophila* (72%). See Figure 8.

### 3.4. Legionella Data Set 3

This data set comprised 553,159 water samples in total, submitted by a range of clients for routine *Legionella* testing. Of these, 60,075 samples were identified as being positive for *Legionella* (11% of all samples). The most prevalent species identified in these samples was *L. anisa* (32,517; around 54% of all positive samples, or 6% of all samples). This included 202 samples which yielded *L. anisa*, as well as other species. The next most prevalent species identified was *L. pneumophila* (17,617; 30% of all positive samples, or 3% of all samples). In a further 4162 samples, other *Legionella* species were identified (7% of all positive samples, or <1% of all samples). This is presented in Figure 9 and Table 4.

## 4. Discussion

The purpose of this study was to highlight that *L. anisa* is an underappreciated opportunistic pathogen in water systems. Through unprecedented access to data made available from three *Legionella* testing laboratories, it was possible to perform an analysis on a total of 565,750 water samples tested for *Legionella*. One data set also provided a species breakdown of *Legionella*-positive colony counts, totaling over 630,000. Increased use of MALDI-ToF MS for the identification of bacterial isolates from culture plates provides more information on speciation than previously available. The fairly recent routine adoption of MALDI-ToF MS by water-testing laboratories means that this is probably the first time that *Legionella* speciation data, on this scale of sample numbers, have been made available for analysis. This enhanced capability to identify *L. anisa* among non-*L. pneumophila* species is reflected in the data examined, highlighting that that it is the most dominant of the non-*L. pneumophila* species. Consequently, analyzing only for *L. pneumophila* could underestimate risk in some circumstances.

The overall picture from the three data sets of routine *Legionella* monitoring of water samples taken from the built environment was that samples positive for *L. anisa* were found to be in a similar proportion to those positive for *L. pneumophila* in one data set, and in a much greater proportion in another. The results may also support previous hypotheses [2,23] that the presence of a more abundant *Legionella* population, including *L. anisa*, in a water system may limit the detection of *L. pneumophila* and that there was a correlation between *L. pneumophila* Sg1 and *L. anisa*, such that an increase in one of the two species led to a decrease in the other, displaying an antagonistic relationship [23,28].

The results supported the previously reported view that *L. anisa* is a common colonizer of water distribution systems and the most frequently identified non-*L. pneumophila* species [21,23,33,34].

Some differences in the proportions of *L. anisa* compared to *L. pneumophila* were found in the three databases studied. In Data Set 1, over a 26-month period, the general proportion of *L. anisa* compared to *L. pneumophila* remained similar at 46% and 47%, respectively. Where this differed, most noticeably in April and May 2020, the proportion of *L. pneumophila*-positive samples increased to 63% and 70%, respectively. This may have been a result of samples taken after a period of premises being shut down following the UK COVID-19 lockdown, suggesting that static water supplies might favor the growth of *L. pneumophila*. Although a smaller total number of samples were submitted for analysis in April 2020, this should not have influenced the proportions of *L. anisa* and *L. pneumophila*.

Although Data Set 2 spanned the period before and after COVID-19 lockdown and indicated a large shift in proportion toward *L. pneumophila* post-COVID, the greatest influence was more likely to have been the intervention of a hospital site-wide disinfection of all hot- and cold-water systems with hydrogen peroxide, and subsequent re-colonization. Although sampling locations where no *Legionella* species were detected had increased from 115 pre-cleaning to 161 post-cleaning, where re-colonization was detected post-cleaning, it appeared to favor *L. pneumophila*. *L. anisa*-positive samples went down from 61 to 38, while *L. pneumophila*-positive samples only fell from 81 to 71. However, examining *Legionella* colony counts showed that the number of *L. anisa* colonies increased as a proportion of the total from 17.2% to 26.4% post-cleaning, while the proportion of *L. pneumophila* was reduced from 73.8% to 64.1% of the total. The total aggregated colony counts had, however, reduced from 437,204 pre-cleaning to 193,869 post-cleaning. A previous study [50] demonstrated a multifactorial relationship between recolonization with different *Legionella* species post-disinfection, including the involvement of biofilms. Whilst TMVs are complex devices with many parts, disinfection with hydrogen peroxide may have been more effective on TMVs than on shower hoses. In the UK, TMVs in hospital settings such as this would all be approved under the Water Regulations Advisory Scheme [58] and have a safety-critical anti-scald function. The same may not be true for all shower hoses, which can be an excellent biofilm-growth environment [59].

Data Set 3, by recording the total number of water samples tested, as well as those found to be *Legionella*-positive, gave us the insight that the positivity rate was around 10%. However, in these samples, *L. anisa* predominated over *L. pneumophila*, 54.1% of the total compared to 29.3%.

## 5. Conclusions

While *L. pneumophila* remains the main *Legionella* species of concern, as it is a risk to human health, the role of *L. anisa* should not be underestimated, either as a potential infection risk or as an indicator of the need to intervene to control *Legionella* colonization of water supplies. There was some evidence that *L. anisa* colonization of water supplies may be more sensitive to environmental influences. For example, *L. pneumophila* may have flourished more in water supplies left static over a shutdown period, and it may have recolonized water supplies after disinfection; however, that may differ according to sample type. There was some indication in a hospital water supply that showers may be more dynamic than TMVs in terms of the reduction in both *L. pneumophila* and *L. anisa* post-cleaning, but also in terms of recolonization. As the evidence from these data sets suggests, *L. anisa*’s colonization of water systems is more prevalent in the UK than previously thought. Consequently, public health officials dealing with clinical specimens should be encouraged to consider other screening methods, in addition to UAT, to detect other *Legionella* species.

## Figures and Tables

**Figure 1 ijerph-21-01101-f001:**
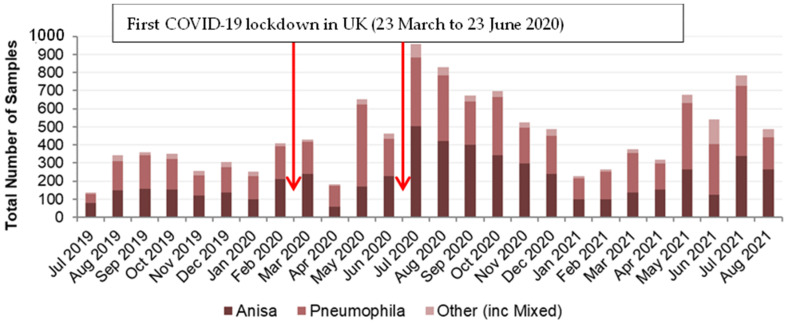
Data Set 1—Total number of *Legionella*-positive samples recorded, categorized by date and species.

**Figure 2 ijerph-21-01101-f002:**
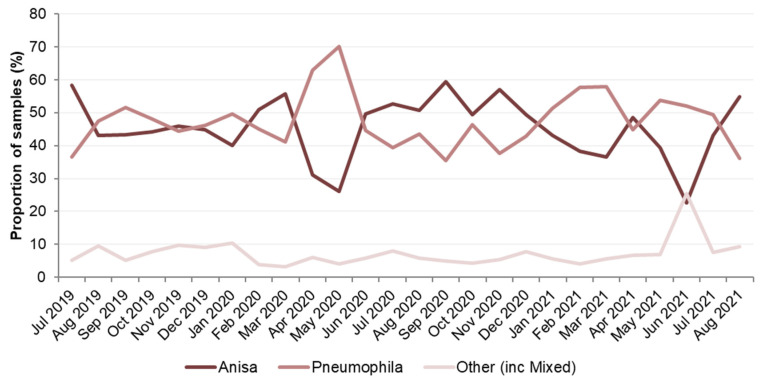
Data Set 1—Proportion of *Legionella*-positive samples recorded, categorized by date and species.

**Figure 3 ijerph-21-01101-f003:**
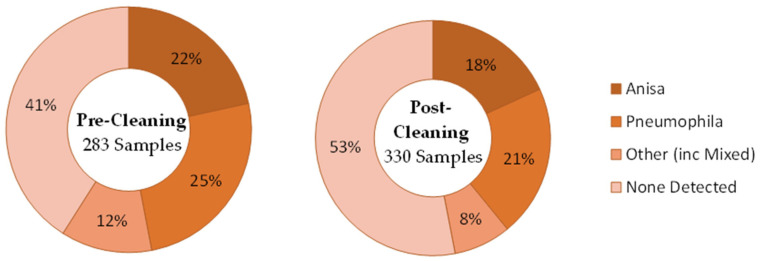
Data Set 2—Total number of *Legionella*-positive samples recorded, categorized by species, for pre- and post-cleaning.

**Figure 4 ijerph-21-01101-f004:**
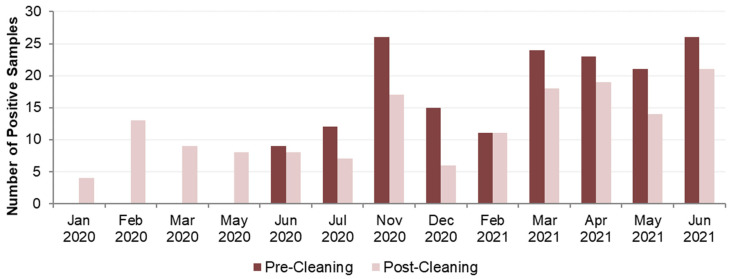
Data Set 2—Total number of *Legionella*-positive samples recorded, categorized by date, for pre- and post-cleaning.

**Figure 5 ijerph-21-01101-f005:**
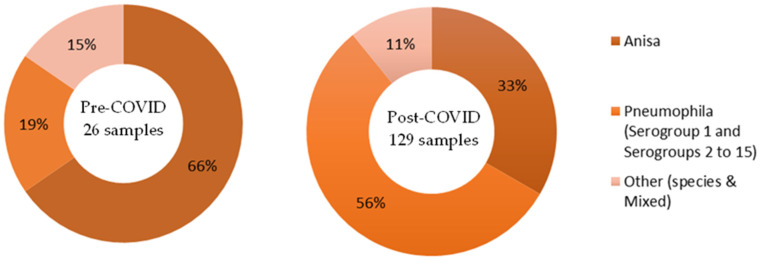
Data Set 2—Number of *Legionella*-positive samples recorded before and after the COVID-19 outbreak, categorized by species.

**Figure 6 ijerph-21-01101-f006:**
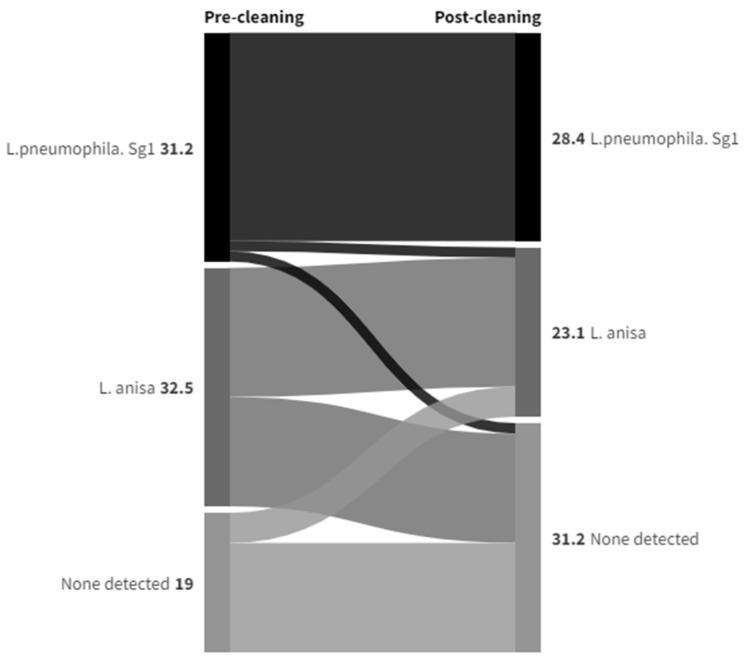
Data Set 2—Changes in species isolated pre- and post-cleaning in shower hoses (numbers = percentage of samples positive for *L. pneumophila* or *L. anisa*, or none detected (other *Legionella* species not relevant to this analysis were excluded).

**Figure 7 ijerph-21-01101-f007:**
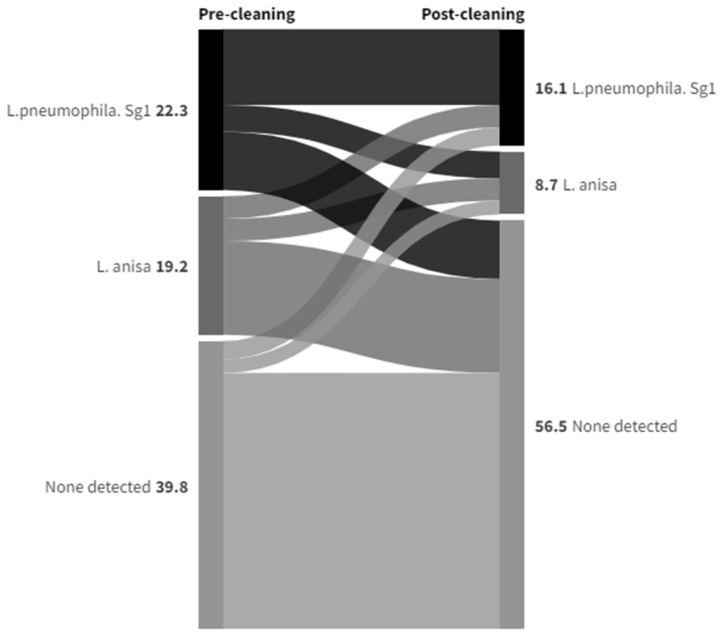
Data Set 2—Changes in species isolated pre- and post-cleaning in TMVs (numbers = percentage of samples positive for *L. pneumophila* or *L. anisa*, or none detected (other *Legionella* species not relevant to this analysis were excluded).

**Figure 8 ijerph-21-01101-f008:**
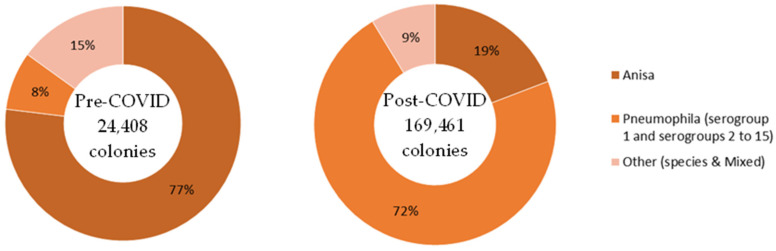
Data Set 2—Number of *Legionella*-positive samples recorded before and after the COVID-19 outbreak, categorized by species.

**Figure 9 ijerph-21-01101-f009:**
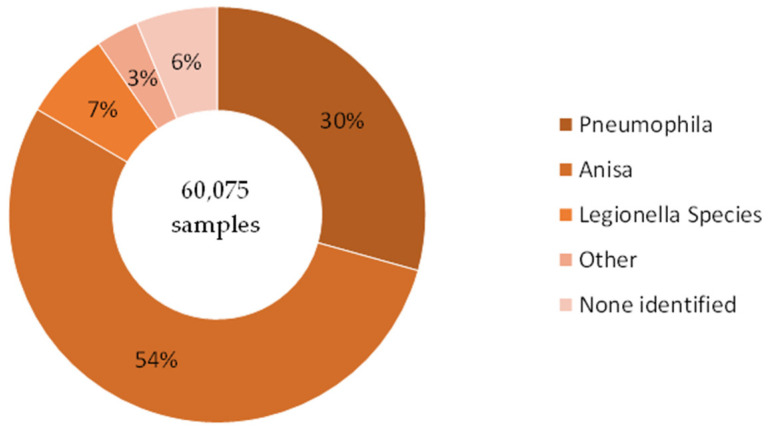
Data Set 3—Total number of *Legionella*-positive samples recorded, categorized by species.

**Table 1 ijerph-21-01101-t001:** Summary of evidence of the contribution of *L. anisa* to contamination of water systems and its clinical significance.

Parameter	Summary	References
*L. anisa* as a colonizer of water distribution systems	After *L. pneumophila*, *L. anisa* is the species most frequently detected in the environment, including from both hot water systems and cooling-tower systems	[14,21,22,23,24,25,26,27,28,29]
	Case reports of colonization of cooling towers, hotel decorative fountain, and restaurant water feature	[5,6,30,31]
	Colonization in cooling towers associated with protozoan Acanthamoeba	[5,32]
	9/20 (45%) hospital water systems in 14 US states colonized with *L. anisa*—sample positivity rate of 57/674 (8.5%)	[33]
	29/108 (27%) electronic tap-water samples are *L. anisa*-positive compared to 25/108 (23%) *L. pneumophila*-positive (USA)	[34]
	Japanese studies >40% samples from public buildings contaminated with *L. pneumophila* and *L. anisa*; *L. anisa* persisted in hospital water for at least 6 years, causing sporadic contamination of shower heads	[25,35]
Frequency of environmental detection of *L. anisa*	13.8% of environmental water sample isolates, but only 0.8% of clinical isolates	[21]
	10/8066 (<0.1%) confirmed clinical cases of legionellosis in New Zealand, Australia, Europe, USA, and Japan attributable to *L. anisa*	[9]
	In 508 community-acquired pneumonia cases, 0.2% cases were attributable to *L. anisa*, and 91.5% to *L. pneumophila*	[17]
Non-pneumonic conditions attributable to *L. anisa* infection	Chronic endocarditis, pleural infection, osteomyelitis, and mycotic aortic aneurysm	[11,36,37,38]
Clinical testing for *L. anisa* by urinary antigen tests (UATs)	UAT-sensitive and -specific but target *L. pneumophila* Sg1, possibly missing up to 40% of cases, and limited for matching environmental samples	[2,9,33,39,40,41]
Clinical testing for *L. anisa* by polymerase chain reaction (PCR)	>80% LD cases in Denmark diagnosed by PCR—likely to increase detection in non-Sg1 cases; thus, non-*L. pneumophila* species found to cause at least 4.5% of all LD cases	[22,42]
	Spanish study of 210 urine samples from suspected pneumonia, and only 4 (1.9%) tested positive for Sg1 by UAT, while by PCR, these 4 tested positive for Sg1 plus 10 more positive for unidentifiable Legionella species and 1 positive for *L. anisa*	[8]
	In a Colombian study of HIV-associated pneumonia patients, 17 had Legionella co-infection—all culture negative, but 6/17 PCR positive for *L. anisa* and 3/17 for *L. pneumophila*	[43]

**Table 2 ijerph-21-01101-t002:** Data Set 2—The effect of cleaning on the species identified in the *Legionella*-positive samples.

		Post-Cleaning Sample				Grand Total
		Anisa	Pneumophila	Other	ND	
**Pre-Cleaning Sample**	**Anisa**	18	6	3	34	**61**
	**Pneumophila**	8	54	4	15	**81**
	**Other**	6	6	4	9	**25**
	**None Detected**	6	5	1	103	**116**
**Grand Total**		**38**	**71**	**12**	**162**	**283**

**Table 3 ijerph-21-01101-t003:** Data Set 2—Total aggregated number of *Legionella*-positive colonies recorded, categorized by date and species, for pre- and post-cleaning.

Date	Pre-Cleaning Sample Colonies				Post-Cleaning Sample Colonies			
	Anisa	Pneumophila	Other	Total	Anisa	Pneumophila	Other	Total
Jan 2020					40	1520	0	**1560**
Feb 2020					12,920	80	1268	**14,268**
Mar 2020					5820	360	2400	**8580**
May 2020					2920	2120	1460	**6500**
Jun 2020	4762	7630	0	**12,392**	3081	1540	0	**4621**
Jul 2020	3180	460	6400	**10,040**	3340	0	180	**3520**
Nov 2020	11,080	25,080	4200	**40,360**	1260	10,820	1400	**13,480**
Dec 2020	2520	8280	0	**10,800**	120	3180	0	**3300**
Feb 2021	2821	26,496	0	**29,317**	408	19,635	4400	**24,443**
Mar 2021	11,673	50,127	2120	**63,920**	8937	37,380	0	**46,317**
Apr 2021	28,875	39,480	17,260	**85,615**	940	11,540	3860	**16,340**
May 2021	3400	111,440	0	**114,840**	0	20,880	200	**21,080**
Jun 2021	6680	53,540	9700	**69,920**	11,460	15,240	3160	**29,860**
**Total**	**74,991**	**322,533**	**39,680**	**437,204**	**51,246**	**124,295**	**18,328**	**193,869**

**Table 4 ijerph-21-01101-t004:** Data Set 3—Total number of *Legionella*-positive samples recorded, categorized by species.

Species Detected	Count	(Proportion)
*Legionella pneumophila*	17,617	(29.3%)
*Legionella anisa*	32,517	(54.1%)
*Legionella* species	4162	(6.9%)
Other	1992	(3.3%)
Blank (no species identified)	3787	(6.3%)
**Total**	**60,075**	

## Data Availability

Data supporting reported results can be obtained upon request from the corresponding author.

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
