# Peer review of "The Contribution of Legionella anisa to Legionella Contamination of Water in the Built Environment"

_ijerph, 2024, doi:10.3390/ijerph21081101_

Round 1

Reviewer 1 Report

Comments and Suggestions for Authors

The Authors present very interesting data on Legionella prevalence, demostrating the role of other species respect with the L.pneumophila.

I have only some points to discuss, that I'm asking to improve:

- Regarding the culture tecnique used, the laboratories have applied the same standard methods?Two different standard methods have been reported, can you described the differences?

- Can the Authors discuss about the type of environmnetal monitoring used from their country? it is self-assestment or istitutional monitoring?

- The Legionella concentration reported need to be considered as CUF/L or others.

- The pre cleaning and post cleaning monitoring needs to be better discuss: how many daysbetween the two activity?

- Figure 6 and 7, releated to Sankey diagrams, need to be discussed, considering that is an unusual way to show changing in bacteria prevalence.

-Table S3, need to be better explained or report in a more clear version;

-Lines 411: The Maldi-tof is able to identify several species, I dont' belive that the identification of more samples containing L.anisa is associated to the capacity of Maldi, but it could be explained with the prevalence of this psecies. I believe that the data on Legionella general prevalence could be reported also in discussion.

- the role of disinfection or effect of TMV on  Legionella need to be speculated.

Im confident in your responce.

Reviewer 2 Report

Comments and Suggestions for Authors

The authors present an article on a very topical and interesting public health issue. Today, it is very useful to understand the role of non-pneumophila Legionella in the contamination of artificial water supplies in order to determine whether there is a risk of Legionnaires' disease for healthy and/or susceptible users. L. anisa is one of the most discussed and problematic environmental species. Therefore, the authors need to better specify the purpose and methodology used in this study.

INTRODUCTION. It is too long, often repeats the same concepts and distracts the reader. It is recommended that the content of this section be streamlined and focused on the key issues of interest for L.anisa.

Materials and Methods.

In this section it is very important to describe well the methodology used both for sampling and for isolation and identification of Legionella. At what time of day (morning or afternoon) and from which sites was the water sample taken: tap and/or shower? hot water, cold water or both? What was the temperature of the water at the time of sampling? Was there a flush prior the sampling? Unfortunately, these are all parameters that greatly influence the results, especially when it comes to L.anisa. If you don't know them, you can't make a valid comment on the results.

Another critical aspect of the study is the evaluation of the results in terms of  number and characteristics of non-comparable samples, especially when carried out on different days. It is well known that the Legionella load is strongly influenced by the frequency of use of the water supply (different between morning and afternoon/evening) and by the external temperature (and therefore by the summer or autumn season). Furthermore, from which territory do the samples examined come? The origin of the water supply is the same for all. L.anisa and L.pneumophila are sensitive to all these conditions. It is advisable to re-evaluate the data considering the same sampling methods, at least in the same months. For example (but it's just an example), immediately before and immediately after the closure due to the lockdown.

RESULTS. They will need to be revised according to the changes that the authors will make in the Materials and Methods section.

DISCUSSION. The work is interesting, but it lacks a more coherent approach with the purpose of the study. Today, epidemiologic data are not useful if they are based only on results obtained from surveillance. Researchers try to understand the reason or conditions that led to these results. It would be very interesting to trace how the reduction in the use of a water network (e.g. during the lockdown period) can increase the risk of legionellosis (see article by De Giglio O, et al. Impact of lockdown on the microbiological status of the Hospital network water network over COVID-19). environmental research 2020:110231 doi: 10.1016/j.envres.2020.110231). Or how L. anisa is not always correctly identified by all laboratories when the identification methods are not the most appropriate, leading to underestimated diagnoses. Or how water temperature affects the replication or survival of L. anisa compared to L. pneumophila.

There are many scientific articles dealing with these aspects. Consider them and the work reported here will become more interesting.

References 

 They can be updated with articles on the influence of closure or poor use of the water network, water temperature and climate change on the colonization of water mains by the genus Legionella.

Reviewer 3 Report

Comments and Suggestions for Authors

-English should be reviewed.

 Line 131: Most probable number is a count method, it is not use for identification.

Viability qPCR methods should be mentioned.

Line 182: please specify the country.

 Through the UK trade body the Legionella Control Association: please review this construct

 2. Materials and Methods: please specify the method of detection: GVPC or BCYE, incubation conditions, sample collection, sample seeding (direct, filtration, volume, etc.).

 Section 1.1.. must be 3.1?

“As this study relied on the goodwill of the laboratories to participate”: it sounds unscientific

State the total number of samples.

 For the third data set, the results 210 provided were not time-resolved but were from a similar time frame. it sounds unscientific

Highlight the months with high detection, for example Jul 2020

There were some months with large differences, most noticeably April and May 2020 and June 2021. In Figure 1

“with a dip in the percentatge of samples identified as L. anisa”: do you mean a “decrease”?

In Figure 1, an arrow indicates when the UK COVID-19 lockdown began, more 243 specifically from 23rd March 2020. You should also mark in figure 1 the ending of this COVID-19 lockdown period.

However, the proportion of those 254 samples identified as L. anisa did not alter. Was not altered

 A Chi-square test indicates that there is no sig-256 nificant association between Legionella species and Pre/Post-COVID (p=0.467). Detection of Legionella globably seems to be increased from Mar 2020. Could you compare these data (before and after in total detection)?.

 intervention and intensive sampling exercise: what do you mean??

Figure 3: ND: what do you mean?

Figure 5: SG and SPP: what is the meaning?

Table 2: if you only do identification of a colony. Is it possible to attribute all colonies in the plate to the sample species? Is a statistically comparison possible?

Figure 6 and 7: legend should express the quantitative data (31.2, 23.1, etc.).

 A greater proportion of TMVs than showers showed no presence of Legionella both 356 pre- and post-cleaning (35.4% compared to 14.9%) and also more changed from having L. 357 pneumophila Sg1 colony counts pre-clean to None Detected post-clean (8.1% compared to 358 1.4%). Is it possible a statistical comparison?

Overall 13% of Legionella-positive sample colonies were identified pre- 374 COVID and 87% post-COVID. The majority of colonies identified pre-COVID were L. 375 anisa (72%), however, post-COVID the majority of colonies were L. pneumophila (77%). How do you explain these changes? Is it possible a statistical comparison?

Table 3: Is it possible a statistical comparison?

 Although sampling locations where no Legionella species were detected had in-427 creased from 115 pre-cleaning to 161 post-cleaning, where re-colonization was detected 428 post-cleaning it appeared to favour L. pneumophila. How do you explain this shift?

In the discussion section, bibliographical data should be provide to explain the findings obtained in the study.

The overall picture from three data sets of routine Legionella monitoring of water sam-444 ples taken from the built environment was that, while the previous focus has tended to be 445 on L. pneumophila, it is clear that L. anisa is also a major contributor. Samples positive for 446 L. anisa were found in a similar proportion to those positive for L. pneumophila in one data 447 set, and in a much greater proportion in another. The results may also support previous 448 hypotheses (Mulder and Yu 2002, van der Mee-Marquet, et al 2005) that the presence of a 449 more abundant Legionella population, including L. anisa, in a water system may limit the 450 detection of L. pneumophila. This is discussion. 

A common section of results and discussion could be added and a conclusion section, with only one paragraph, with the main findings. 

Comments on the Quality of English Language

English should be improved, with the review of an English speaking professional. 

Reviewer 4 Report

Comments and Suggestions for Authors

Review of the manuscript ijerph-3111805 entitlet “The contribution of Legionella anisa to Legionella contamination of water in the built environment”. In this manuscript authors reported results of different dataset in order to evaluate the relationship between the presence of L. anisa and L. pneumophila. The topic is interesting. However the manuscript showed some critical issues and should be further improved prior to publication. In particular i) the introduction is too long and should be reorganized; ii) discussion should be improved because is very poor; iii) the conclusions should be supported by the results obtained and they should not contain part of the discussion. Moreover a reading of the manuscript by a native english speaker is suggested.

Other remarks:

Introduction

 Page 2 Lines 90-92 – This sentence seems in contrast with the previous one.

Page 4 Lines 182-185 – The aim of the study should be better clarified.

Materials and methods

Page 4 Lines 191-194 – The methods used for Legionella detection should be briefly described.

Results

Page 8 Line 311– For the 630,000 value the measure unit should be reported (CFU/L?).

Page 9 – It would be better to compare the CFU before and after treatment for a single sample rather than in total

Page 9 – Lines 324-330 and 331-341 should be deleted. Only the general comment and the Figure should be reported.

Page 11 Line 366 – Showers were recolonized by… ? (L. anisa o L. pneumophila?).

Page 11 Lines 373-377 – These considerations should be deleted because the results are influenced by the treatment.

Discussion

Page 11 Lines 406-407 – Delete, see the previous comment (Page 9).

Page 12 Lines 410-413 – These data are not reported in the results section.

Page 12 Line 416 – Similar? Respect to what?

Page 13 Lines 423-427 – Delete, see the previous comment (Page 11).

Conclusion

Page 13 Lines 444-446 – This sentence seems not supported by the data.

Page 13 Lines 449-451 – This sentence is not clear.

Table 1 – The total samples should be 283 basing on that reported in page 6 line 266.

Table 2 – For the different values the measure unit should be reported (CFU/L?).

Table 3 – For the different values the measure unit should be reported (CFU/L?).

Comments on the Quality of English Language

A reading of the manuscript by a native english speaker is suggested.
